# Fungicidal Activity of Zinc Oxide Nanoparticles against Azole-Resistant *Aspergillus flavus* Isolated from Yellow and White Maize

**DOI:** 10.3390/molecules28020711

**Published:** 2023-01-11

**Authors:** Nuha M. Alhazmi, Eman M. Sharaf

**Affiliations:** 1Department of Biology, College of Science, University of Jeddah, Jeddah 21589, Saudi Arabia; 2Department of Bacteriology, Immunology, and Mycology, Animal Health Research Institute (AHRI), Shebin El Kom 32511, Egypt

**Keywords:** zinc oxide nanoparticle, azole-resistant *Aspergillus flavus*, yellow maize, white maize and antifungal agent

## Abstract

The risk of resistance development and adverse effects on human health and the environment has increased in the last decade. Furthermore, many antifungal agents fail to inhibit the pathogenesis of azole-resistant *Aspergillus flavus*. In this report, we isolated and identified azole-resistant *A. flavus* isolates from two sources of maize (white and yellow maize). The susceptibilities of *Aspergillus flavus* isolates were investigated by conventional antifungals such as Terbinfine, Fluconazole, Ketoconazole, Voricazole, Amphotericin, and Nystatin. Then zinc oxide nanoparticles associated with *Chlorella vulgaris*, which are synthesized by using the precipitation method, were examined against isolated fungi. The results showed that twelve species of white corn were isolated out of fifty isolates, while the number of isolates from the yellow corn source was only four. Interestingly, the following antifungals have an impact effect against azole-resistant *A. flavus* isolates: the inhibition zones of ketoconazole, voricazole, and terbinafine were 40 mm, 20 mm, and 12 mm, respectively, while the remaining antifungal agents have no effect. Similarly, the inhibition zones of the following antifungal agents were as follows: 41 mm for Terbinfine, 13 mm for Voricazole, and 11 mm for Ketoconazole against *Aspergillus flavus* that was isolated from yellow corn. The physiochemical characterization of zinc oxide nanoparticles provides evidence that ZnO-NPs associate with *Chlorella vulgaris* and have been fabricated by the precipitation method with a diameter of 25 nm. The zinc oxide nanoparticle was then used to isolate azole-resistant *A. flavus*, and the results show that ZnO-NPs have an effect on azole-resistant *A. flavus* isolation. The inhibition zone of zinc oxide nanoparticles against *A. flavus* (that was isolated from white corn) was 50 mm with an MIC of 50 mg/mL, while the inhibition zone of zinc oxide nanoparticles against Azole-resistant *A. flavus* isolated from yellow corn was 14 nm with an MIC of 25 mg/mL, which indicated that zinc oxide nanoparticles gave a better result against Azole-resistant *A. flavus* isolated from maize.

## 1. Introduction

Plant pathogens may lead to a reduction in the yield as well as the quality of agricultural crops, leading to substantial economic loss [1,2,3,4]. Previous studies reported that crop losses for five of the commodities most widely grown—wheat, rice, maize, potatoes, and soybeans—were in the range of 20 to 30 percent. Food security and “zero hunger” are among the global goals declared by the United Nations [5]. The scientists’ efforts to reduce fungal diseases are crucial for guaranteeing sufficient, high-quality food for a growing world population. Modern agriculture has become increasingly reliant on the application of fungicides to mitigate fungi-related crop losses and fulfill this goal. Fungicides include a broad range of compounds with different modes of action that belong to various chemical classes. The extensive use of chemical pesticides in agriculture also raises concerns for public health. Exposure experiments in rats showed endocrine-disrupting, biochemical, histopathological, and hematological effects [6]. Several fungicides are classified as hazardous chemicals by the World Health Organization (WHO) and are banned in the European Union. The use of azoles for non-medical purposes has a wide range, from agriculture and horticulture to the prevention of post-harvest losses [7]. Azoles target a broad spectrum of fungal pathogens in various crops. *A. flavus* is a plant pathogen that is able to infect and colonize many crops, such as corn and cotton. It causes an economic problem due to the infection of strategic crops such as maize [4]. *A. flavus* can cause subsequent contamination of the grain with the fungal metabolite aflatoxin. Aflatoxin-classified polyketide-derived furanocoumarins are produced by the genus Aspergillus and are classified as highly carcinogenic and toxic materials for humans and livestock [5,6]. Maize is an economic crop that provides nutrients for humans and livestock. It cultivates in many tropical areas, such as North America, Africa, and the Middle East. It includes three types (red, yellow, and white) [5]. Nanotechnology contributes to many different fields, such as medicine, industry, and agriculture. Nowadays, researchers are focusing on the benefits of nanomaterials, such as their large surfaces and their optical and chemical properties. Nanomaterials include different metals, such as noble metals and metal oxides [6]. Zinc oxide has been applied in the pharmaceutical and cosmetic industries in powder and ointment forms. Furthermore, it has been successful as an antimicrobial, antifungal, and anticancer drug [7,8]. In this study, we isolated *A. flavus* from 100 samples of maize grains (fifty samples from white maize and fifty samples from yellow maize) and then investigated the triazole susceptibility of isolates against antifungal drugs, determined the MIC of the collected isolates, and, finally, utilized zinc oxide nanoparticles as fungicides against azole-resistant *A. flavus* isolates.

## 2. Results 

### 2.1. Isolation and Identification of Azole-Resistant A. flavus

According to Table 1, the total number of azole-resistant *A. flavus* isolates was 12 out of 50 samples of white maize, while 4 isolates of azole-resistant *A. flavus* were isolated from yellow maize. Figure 1 and Figure 2 show the azole resistance of *A. flavus* in white and yellow maize, respectively.

### 2.2. Characterization of Zinc Oxide Nanoparticle

As Figure 3 shows, there was a broad band at 3477 cm^−1^ corresponding to the hydroxyl group of the water molecule on the surface of the ZnO nanoparticle. The band at 1551 cm^−1^ is due to an OH bend to ZnO. A strong band at 478 cm^−1^ was also attributed to Zn-O. XRD patterns of ZnO nanoparticles are shown in Figure 4. The peaks at 2 θ = 31.746, 34.395, 36.226, 47.526, 56.549, 62.832, 67.893, and 69.028 were assigned to (100), (002), (101), (110), (103), (200), (112), and (201) of ZnO nanoparticles. All peaks indicated a hexagon Wurtzite structure (Zincite, JCPDS no.: 89-0510). No characteristic peaks of any impurities were detected, demonstrating the high quality of the ZnO nanoparticle. The SEM image in Figure 5 displayed the well-defined shape of zinc oxide nanoparticles with a smooth surface. Furthermore, the TEM image (Figure 6) showed that the morphology and shape were like rods, and its size was estimated at 25–30 nm.

### 2.3. The Susceptibility of Aspergillus flavus Isolates against Antifungal Agents

As shown in Table 2 and Figure 7A,B, the antifungal susceptibility of all isolates investigated against different antifungal agents is good; the common antifungal agents such as Terbinfine, Voricazole, and Ketoconazole have good antifungal activity against *A. flavus* that was isolated from white and yellow maize as follows: Ketoconazole > Voricazole > Terbinfin, with inhibition zones of 40 mm, 20 mm, and 12 mm for Terbinfin, Voricazole, and Ketoconazole, respectively, for *Aspergillus flavus* isolated from white corn. Similarly, the inhibition zones of the following antifungal agents were as follows: 41 mm for Terbinfine, 13 mm for Voricazole, and 11 mm for Ketoconazole against *A. flavus* that was not isolated from yellow corn. However, the rest of the antifungal agents have not had any effect against *A*. *flavus,* which was isolated from both white and yellow maize.

### 2.4. The Susceptibility of Aspergillus flavus Isolates against Zinc Oxide Nanoparticles

Zinc oxide nanoparticles have good antifungal activity against many funguses, such as *Candida albicans*, *Aspergillus niger,* and *Pestalotiopsis maculans* [9], as Table 3 and Figure 8A,B reveal. Zinc oxide nanoparticles which are synthesized by association with *Chlorella vulgaris,* have a strong effect against both isolates of *A. flavus* with azole resistance. The inhibition zone of *A. flavus* isolated from white maize is 14 nm, while, zinc oxide nanoparticles have less effect on *A. flavus* isolated from yellow maize. Similarly, the minimal inhibitory concentration of zinc oxide nanoparticles against Azole-resistant *A. flavus* isolated from yellow corn (25 vs. 50 mg/mL) is higher than that of white corn.

## 3. Discussion

In the past two decades, fungal infections have been growing around the world. This is because the number of patients who have received hematopoietic stem cell or solid organ transplantation and immunosuppressive therapy has increased. Fungal infection diseases still represent a great threat to individuals’ health, although novel antifungals have been developed [10]. The most common fungal infections may be caused by Aspergillus species that lead to aspergillosis. Several studies focus on genes responsible for gene resistance, virulence factors yielded by *A. flavus* during the colonization of maize tissues, as well as maize kernel virulence proteins [11]. Based upon the proteomics tools to identify and characterize the virulence proteins produced by A. flavus, it includes hydrolytic enzymes instead of amylases: cellulases, chitinases, cutinases, lipases, and pectinases (P2c). Researchers are interested in the cyp51A alleles that exist in azole-resistant isolates from the environment. The majority of the resistant genes in resistant isolates were found in either the TR34/L98H allele (60.7%) or the TR46/Y121F/T289A allele (15.0%). This allele was found in 29.3% of the azole-resistant environmental isolates in Africa, 37.4% of the resistant isolates in East Asia, 77% of the resistant isolates from Europe, 88.9% of the resistant isolates from India, 37.1% of the resistant isolates from the Middle East, 58.8% of the resistant isolates from North America, and 4.3% of the resistant isolates from South America [12,13]. Zinc oxide nanoparticles are registered as having better biocompatibility than other metal nanoparticles. Many studies provide evidence that the use of ZnO nanoparticles has a negligible potential adverse effect on public health. It has a wide range of applications, such as sunscreens, toothpastes, anti-dandruff shampoos, and anti-fouling paints [14]. Furthermore, zinc oxide nanoparticles have a successful impact as an anticancer agent as well as drug delivery [15,16]. The physical properties of zinc oxide nanoparticles have a central role in their antifungal activity. It includes particle size, concentration, morphology, and superstructure, which affect the function of ZnO nanoparticles as nano additives in food-related products. Instead, Sharma et al. have reported that the inhibitory activity of ZnO-NPs on the fungus Fusarium sp. was concentration-dependent as well as particle-sized [17]. Padmavathy et al. also stated that particle size and concentration had an effect on the antifungal performance of ZnO-NPs [16]. We isolated and identified azole resistance in *A. flavus* from two types of maize grain fields (white and yellow maize) that had been sprayed with azole fungicides and then treated with conventional antifungal drugs in our study and applied zinc oxide nanoparticles associated with Chlorella vulgaris, which are synthesized by using the precipitation method, against isolated fungi. The out-finding data showed that twelve species of white corn were isolated out of fifty isolates, while the number of isolates from the yellow corn source was only four. Interestingly, the inhibition zones of ketoconazole, voriconazole, and terbinafine against isolates of Azole-resistant *A. flavus* were 40 mm, 20 mm, and 12 mm, respectively, whereas the rest of the antifungal agents had no effect. Similarly, the inhibition zones of the following antifungal agents were as follows: 41 mm for Terbinfine, 13 mm for Voricazole, and 11 mm for Ketoconazole against *A. flavus* that was isolated from yellow corn. The physiochemical characterization of zinc oxide nanoparticles provides evidence that ZnO-NPs associate with Chlorella vulgaris and have been fabricated by the precipitation method with a diameter of 25 nm. The zinc oxide nanoparticles were then applied to isolate Azole-resistant A. flavus, and the results show that ZnO-NPs have an effect on isolate Azole-resistant A. flavus. The inhibition zone of zinc oxide nanoparticles against *A. flavus* (that was isolated from white corn) was 50 mm with a MIC of 50 mg/mL, while the inhibition zone of zinc oxide nanoparticles against Azole-resistant *A. flavus* isolated from yellow corn was 14 nm with a MIC of 25 mg/mL, which indicated that zinc oxide nanoparticles gave a better result against Azole-resistant *A. flavus* isolated from yellow corn. The antifungal mechanism of zinc oxide nanoparticles depends on their size and shape [18]. Nanoparticles exist in different shapes, such as rod-shaped or spherical [19,20,21], which have an effect on the surface area to volume ratio and the ability to cause physical damage to cells. The proposed fungicidal mechanisms of nanoparticles are extrapolated from those proposed for bacteria and include interactions with thiol groups of vital enzymes leading to enzyme inactivation [22] and killing by oxidative stress [18,23]. ROS was, however, demonstrated to cause membrane and cell wall damage [24]. Hence, scientists noticed that the addition of nanoparticles can induce the generation of ROS in a time- or dose-dependent manner [24]. Finally, zinc oxide nanoparticles are recommended as antifungal agents against azole-resistant *A. flavus*.

## 4. Material and Methods

### 4.1. Sampling, Isolation, and Identification of Aspergillus flavus

According to Pitt et al. (2009) [25] azole-resistant *A. flavus* isolates were collected from maize grain fields that had been sprayed with azole fungicides. Then, 30 mL of 0.1 M sodium pyrophosphate was added to the collection jar and vortexed vigorously for 30 s. After the vortexing process, the jar was left standing for 1 min, and then 100 μL of the supernatant were each plated on two Sabouraud dextrose agar (SDA) plates supplemented with chloramphenicol (0.05 mg/L) and gentamicin (0.05 mg/L). Plates were incubated at 45 °C for up to 7 days, and colonies were harvested when they were 2–4 mm in diameter. The plated volume of supernatant was adjusted, if necessary, in a subsequent plating for each sample to achieve <10 total colonies per plate. The isolates were preliminarily identified by microscopic morphology [25].

### 4.2. Synthesis of Zinc Oxide Nanoparticles Using Chlorella vulgaris

Zinc acetate dehydrate (0.02 M) was added to 40 mL of distilled water under constant stirring. Ten milliliters of algal aqueous extract was added dropwise to this solution after 10 min of stirring. NaOH (2.0 M) was added until reaching pH 12; the resulting pale white aqueous solution was magnetically stirred for 2 h. The precipitate was washed twice with distal deionized water, followed by ethanol, to obtain a solution free of impurities. The precipitates were dried at 60 °C in a vacuum oven overnight [26].

### 4.3. Characterization of Zinc Oxide Nanoparticles

The characterization of the zinc oxide nanoparticle was performed using the following physiochemical techniques: Fourier transform infrared (FT-IR) spectrum via the Nicolet 6700 apparatus (Thermo Scientific Inc., Waltham, MA, USA ), the crystalline nature and grain size were analyzed by XRD (D8 Advance X-ray Diffractometer, Bruker, Germany), the morphology and visualization of zinc oxide nanoparticles were evaluated by Transmission Electron Microscopy (TEM, JSM-2100F, JEOL Inc., Tokyo, Japan) and Scanning Electron microscopy (SEM, JSM-690, JEOL Inc., Tokyo, Japan).

### 4.4. Antifungal Disk Diffusion Method

The fungal strains were cultured on Sabouraud dextrose agar and incubated at 35 °C for 24 h and for 5 days on potato dextrose agar slant for the mold fungi. Using a sterile loop, pure colonies of the Aspergillus flavus species were transferred into a tube containing sterile normal saline. For the mold, 1 mL of sterile distilled water supplemented with 0.1% Tween 20 was used to cover and resuspend the colonies. Using a hemocytometer, the suspension was adjusted to 2–5 × 10^6^ conidia/mL. The suspension was further diluted by 1:10 to obtain final working inoculums 2–5 × 10^5^ conidia/mL. The inoculums were poured over MHA supplemented with 2% glucose. The sterile 6 mm disks that were impregnated with 20 μL test compound (with a concentration of 10 mg/mL) were placed over the plate. Nystatin, a standard antifungal drug, was used as a positive control, and sterile distilled water was used as a negative control, with the cultures incubated at 35 °C for 48 h. The zone of inhibition was measured in millimeters [27].

### 4.5. The Susceptibility of *Aspergillus flavus* Isolates

The MICs for *A. flavus* were determined using the reference procedures of the Antifungal Susceptibility Testing of CLSI M27-A3 and EUCAST for the testing of fermentative yeasts. MICs for *A. flavus* were determined in accordance with EUCAST and CLSI M38-A. Briefly, testing was performed in sterile 96-well microtiter plates with Roswell Park Memorial Institute (RPMI) 1640 medium with l-glutamine, without sodium bicarbonate (NaHCO3, RPMI 1640; Gibco, Carlsbad, CA, USA), supplemented with 2% glucose, buffered to pH 7.0 with 4-(2-hydroxyethyl)-1-piperazineethanesulfonic acid (HEPES) medium [28,29,30].

### 4.6. Preparation of Fungal Inoculums

Regarding the preparation of yeast inoculums, the fungal strains were subcultured on Sabouraud’s dextrose agar slant and incubated for 24–48 h at 35 °C to obtain a freshly grown pure culture. The homogenous suspension was adjusted to 0.5 McFarland standards. Then, the inoculum size was further adjusted to 0.5 × 10^5^ or 2.5 × 10^5^. In addition, the mold suspension of conidia was obtained after 5 days of culture on Sabor and dextrose agar slants incubated at 35 °C. Colonies were covered with 5 mL of sterile distilled water supplemented with Tween 20. The conidia were collected with a sterile cotton swab, transferred to a sterile tube, and vortexed to homogenize the suspension. The suspension was standardized by counting the conidia in a hemocytometer to 2–5 × 10^6^ conidia/mL. The suspension was diluted 1:10 with RPMI to obtain final inoculums of 2–5 × 10^5^ conidia/mL. A total of 50 μL of each compound concentration and 50 μL of fungal suspension were added to each well for the negative control lane, but 100 μL of broth was added to the positive control lane (each well reached the final desired concentration of 2–5 × 10^5^ CFU/mL). The plate was sealed with aluminum foil and incubated at 35 °C for 24 and 48 h in a humid atmosphere. The MIC was determined using an ELISA reader at 530 nm for the yeast species and visually for the mold species after 48 h of incubation as the lowest concentration of drug that resulted in 50% inhibition of growth compared to drug-free growth control. The list of standard antifungals is listed in Table 4.

### 4.7. The Antifungal Activity of Zinc Oxide Nanoparticle against Azole-Resistant A. flavus Isolates

#### 4.7.1. Antifungal Disk Diffusion Method

The fungal strains were cultured on Sabouraud dextrose agar and incubated at 35 °C for 24 h and for 5 days on potato dextrose agar slant for the mold fungi. Using a sterile loop, pure colonies of the *A. flavus* species were transferred into a tube containing sterile normal saline. For the mold, 1 mL of sterile distilled water supplemented with 0.1% Tween 20 was used to cover and resuspend the colonies. Using a hemocytometer, the suspension was adjusted to 2–5 × 10^6^ conidia/mL. The suspension was further diluted by 1:10 to obtain final working inoculums 2–5 × 10^5^ conidia/mL. The inoculums were poured over MHA supplemented with 2% glucose. The sterile 6 mm disks that were impregnated with 20 μL of zinc oxide nanoparticles were dissolved in dimethyl sulfoxide (DMSO) (with a concentration of 10 mg/mL) and placed over the plate. As a negative control, sterile distilled water was incubated at 35 °C for 48 h. The zone of inhibition was measured in millimeters [27].

#### 4.7.2. Minimum Inhibitory Concentration (MIC)

Regarding the preparation of *A. flavus* inoculums, the fungal strains were subcultured on Sabouraud’s dextrose agar slant and incubated for 24–48 h at 35 °C to obtain a freshly grown pure culture. The homogenous suspension was adjusted to 0.5 McFarland standards. Then, the inoculum size was further adjusted to 0.5 × 10^5^ or 2.5 × 10^5^. In addition, the mold suspension of conidia was obtained after 5 days of culture on Sabor and dextrose agar slants incubated at 35 °C. Colonies were covered with 5 mL of sterile distilled water supplemented with Tween 20. The conidia were collected with a sterile cotton swab, transferred to a sterile tube, and vortexed to homogenize the suspension. The suspension was standardized by counting the conidia in a hemocytometer to 2–5 × 10^6^ conidia/mL. The suspension was diluted 1:10 with RPMI to obtain final inoculums of 2–5 × 10^5^ conidia/mL. A total of 50 μL of zinc oxide nanoparticles was dissolved in dimethyl sulfoxide (DMSO) (with a concentration of 10 mg/mL), and 50 μL of fungal suspension was added to each well for the negative control lane, but 100 μL of broth was added to the positive control lane (each well reached the final desired concentration of 2–5 × 10^5^ CFU/mL). The plate was sealed with aluminum foil and incubated at 35 °C for 24 and 48 h in a humid atmosphere. The MIC was determined using an ELISA reader at 530 nm for *A. flavus* and visually for mold species after 48 h of incubation as the lowest concentration of drug that resulted in 50% inhibition of growth compared to drug-free growth control [31].

## 5. Conclusions

In the study, we isolated *A. flavus* from two types of maize (white and yellow maize) grain fields that had been sprayed with azole fungicides by the microscopic method and examined their resistance to conventional antifungal drugs. Then, investigate the efficiency of zinc oxide nanoparticles associated with *Chlorella vulgaris*, which are synthesized by using the precipitation method, against isolated fungi. The result showed that only 12 species of white corn were isolated out of 50 samples, while the number of isolates from the yellow corn source was only 4. Interestingly, the inhibition zones of the following drugs: ketoconazole, voricazole, and terbinafine have an effect on Azole-resistant *A. flavus* isolates, whereas the rest of the antifungal agents have no effect on Azole-resistant *A. flavus* isolates. The physiochemical characterization of zinc oxide nanoparticles provides evidence that ZnO-NPs associate with *Chlorella vulgaris* and have been fabricated by the precipitation method with a diameter of 25 nm. Then, we applied the zinc oxide nanoparticle against the isolate of azole-resistant *A. flavus,* and the data indicate that the antifungal activity of ZnO-NPs has an impact effect against isolates of azole-resistant *A. flavus.* The inhibition zone of zinc oxide nanoparticles against *A. flavus* (that was isolated from white corn) was 50 nm with a MIC of 50 mg, while the inhibition zone of zinc oxide nanoparticles against azole-resistant *A. flavus* isolated from yellow corn was 14 mm with a MIC of 25 mg/mL, which indicated that zinc oxide nanoparticles give a better result against azole-resistance. *A. flavus* isolated from yellow corn zinc oxide nanoparticles are very active against the azole-resistant *A. flavus.*

## Figures and Tables

**Figure 1 molecules-28-00711-f001:**
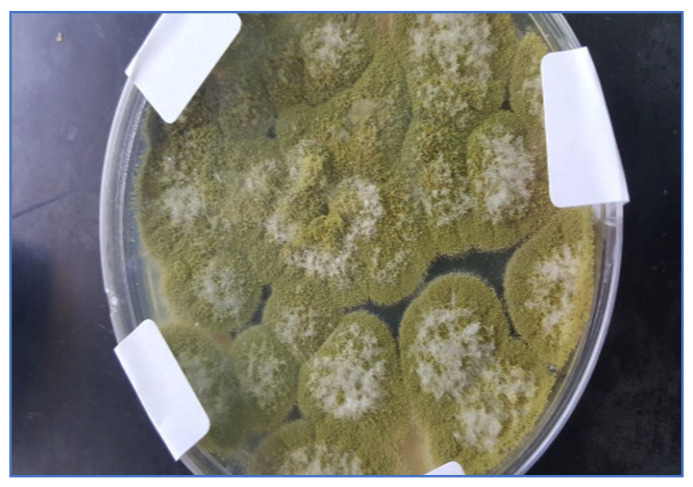
Azole-resistant *A. flavus* isolates, white maize.

**Figure 2 molecules-28-00711-f002:**
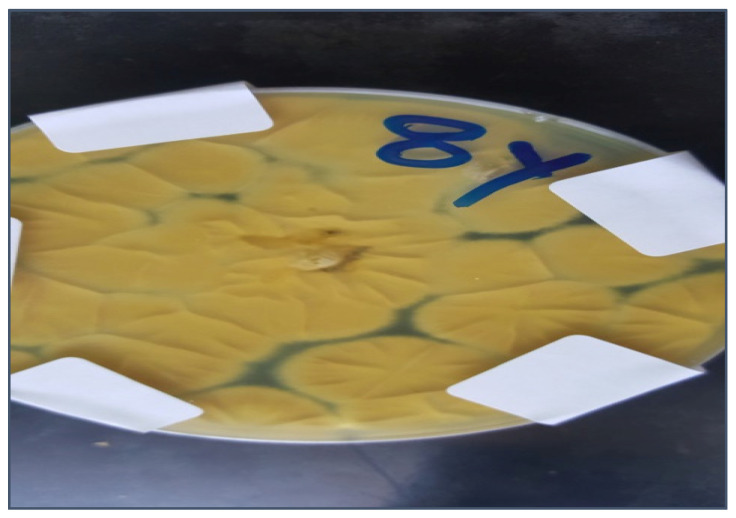
Azole-resistant *A. flavus* isolates, yellow maize.

**Figure 3 molecules-28-00711-f003:**
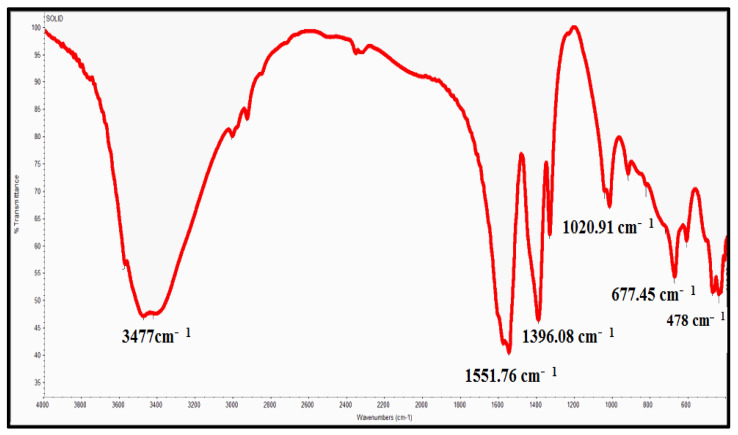
FTIR image of zinc oxide nanoparticles.

**Figure 4 molecules-28-00711-f004:**
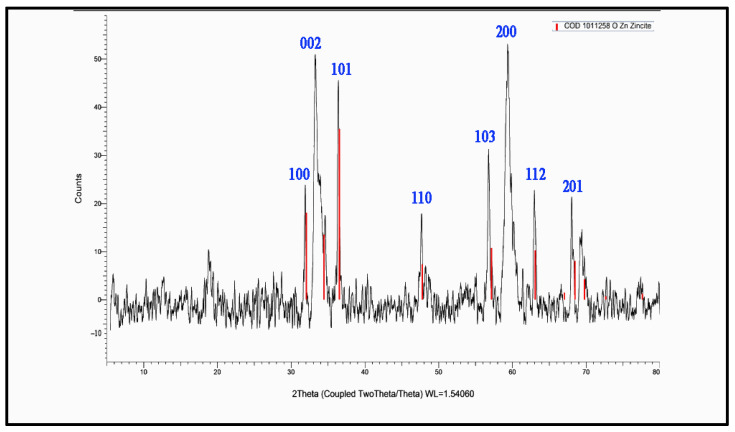
XRD image of zinc oxide nanoparticles.

**Figure 5 molecules-28-00711-f005:**
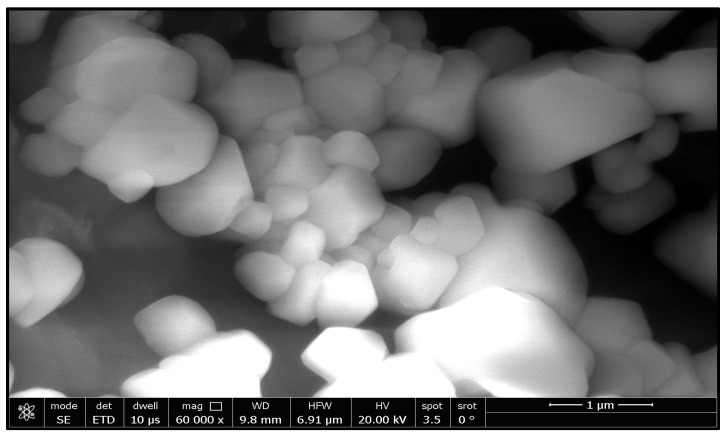
SEM image of zinc oxide nanoparticles.

**Figure 6 molecules-28-00711-f006:**
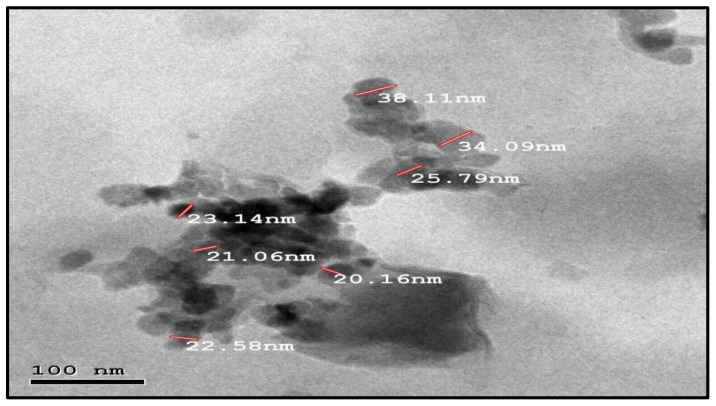
TEM image of zinc oxide nanoparticles.

**Figure 7 molecules-28-00711-f007:**
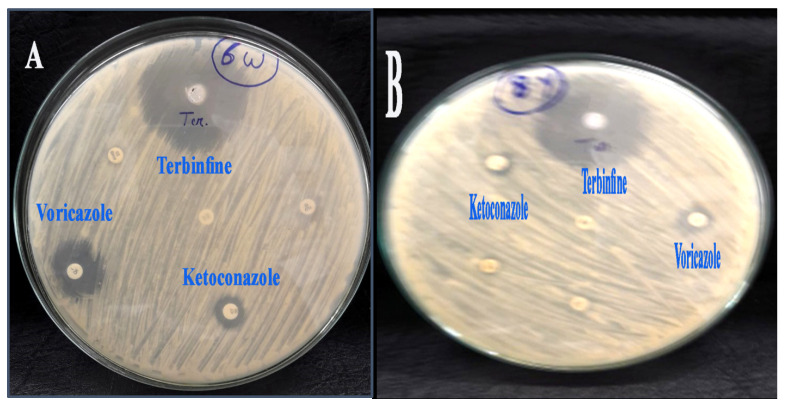
The inhibition zone of antifungal agents against *Aspergillus flavus* isolates from maize. (**A**) *Aspergillus flavus* isolated from white maize. (**B**) *Aspergillus flavus* isolated from yellow maize.

**Figure 8 molecules-28-00711-f008:**
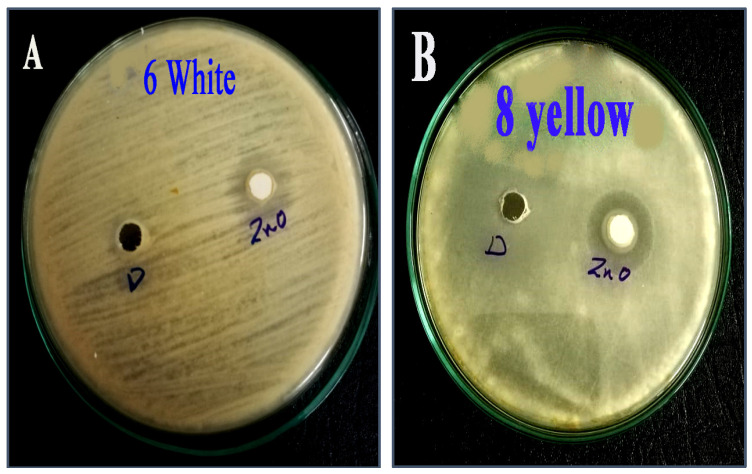
The inhibition zone of zinc oxide nanoparticles against *A. flavus* isolates from maize. (**A**) *A. flavus* isolated from white maize. (**B**) *A. flavus* isolated from yellow maize.

**Table 1 molecules-28-00711-t001:** Number of the *A. flavus* isolates according to the sources.

The Source of Azole-Resistant *A. flavus* Isolates	No. of Isolates
White corn	12
Yellow corn	4

**Table 2 molecules-28-00711-t002:** The inhibition zone of *A. flavus* isolates against antifungal agents.

Tested M.O.	Terbinfine	Voricazole	Ketoconazole	Fluconazole	Amphotericin	Nystatin
*A.flavus* (white corn)	40 ± 0.9	20 ± 0.2	12 ± 0.4	NA	NA	NA
*A.flavus* (yellow corn)	41 ± 0.5	13 ± 0.6	11 ± 0.1	NA	7 ± 0.2	NA

**Table 3 molecules-28-00711-t003:** The inhibition zone of zinc oxide nanoparticles against *A. flavus* isolates from maize.

Tested Micro-Organisms	Inhibition Zone (mm)	MIC (mg/mL)
*A. flavus* (white corn)	13 nm	50 ± 2.3
*A. flavus* (yellow corn)	14 nm	25 ± 1.6

**Table 4 molecules-28-00711-t004:** The antifungal agents with their concentrations.

Antifungal Disc	Concentration
Terbinfine	100 mg/mL
Fluconazole	25 mg/mL
Ketoconazole	30 mg/mL
Voricazole	1 mg/mL
Amphotericin	100 mg/mL
Nystatin	100 mg/mL

## Data Availability

The original contributions presented in the study are included in the article; further inquiries can be directed to the corresponding author.

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
