# Peer review of "Fungicidal Activity of Zinc Oxide Nanoparticles against Azole-Resistant *Aspergillus flavus* Isolated from Yellow and White Maize"

_molecules, 2023, doi:10.3390/molecules28020711_

Round 1

Reviewer 1 Report

I suggest rejecting this manuscript at its current state. The reasons are:

1) It looks like in Abstract the sequence of words has occasionally got changed so that the meaning of the sentences has become totally unclear. There are occasional italics in the text without any reasoning. These serious errors indicate that either there has been a technical error or the authors have no idea what they were writing about.

2) Introduction is very hectic and does not properly introduce the main classes of fungicides or the specific fungicide azole, central to this study, the importance of different types of maize nor the basis for selecting ZnO nanoparticles for the study.

3) The construction of sentences in Materials and Methods section is not correct and the text is very difficult to understand.       

4) In Materials and Methods, it is not totally clear how ZOI and MIC were determined and measured. Why were ZnO particles dissolved in DMSO? Was DMSO used as a negative control?

5) It is unclear what is shown on Figures 1 and 2.

6) It is totally unclear how the authors find based on Figure 5 that the size of ZnO particles was 25 nm. Scale bar on SEM image is 3 micrometers, so the ZnO particles look more around 500-1000 nm in diameter. The TEM image on Figure 6 is of relatively bad quality and indicates that some of the larger particulates are indeed aggregates of ca 25 nm sized nanoparticles.

7) In Table 4, MIC is given in mg, is it mg/L, mg/ml or mg of something else? How could ZOI be measured in nanometers? (e.g, 15 nm=0.015 micrometers=0.000015 millimeters; how can this be measured?) Have the authors calculated any standard deviations?

I suggest the authors significantly improve their manuscript.

Author Response

 POINTWISE RESPONSE TO REVIEWERS’S COMMENTS ON Molecule –MDPI

Dear Molecule –MDPI Editor, Dear Reviewers,

Thank you for the reviewer comments for molecule –MDPI; they are very helpful to improve our manuscript. Please find attached the revised article entitled “Fungicidal activity of zinc oxide nanoparticles against Azole-Resistant Aspergillus  flavus isolated from yellow and white maize ”, for consideration for publication in Molecules- MDPI.

Each reviewer comment is indicated below, together with the responses (clarifications/ revisions) we have included in the revision.

It is hoped that this revision satisfies the queries raised by respected reviewers and manuscript will be considered for publication in Molecules- MDPI.

With kind regards

Dr. Amr Hassan

Reviewer-1

.

  1. It looks like in Abstract the sequence of words has occasionally got changed so that the meaning of the sentences has become totally unclear. There are occasional italics in the text without any reasoning. These serious errors indicate that either there has been a technical error or the authors have no idea what they were writing about

AUTHORS: Thank you very much for your valuable suggestion. Thanks for reviewer’s comments; we fixed all the errors.

  Abstract

Systemic fungicide use has increased over the last decades, despite the susceptibility of resistance development and the side effects to human health and the environment. Many antifungal agents are fail in inhibiting the pathogenesis of Azole-resistant Aspergillus  flavus. In this report, we isolated and identified Azole-resistant A. flavus isolates from two sources of maize (white and yellow maize). The susceptibility of Aspergillus flavus isolates were investigated by convention antifungals such as Terbinfine, Fluconazole, Ketoconazole, Voricazole, Amphotericin and Nystatin. Then applied zinc oxide nanoparticles associated with Chlorella vulgaris, which are synthesized by using the precipitation method, against isolated fungi. The results displayed that that twelve species of white corn were isolated out of fifty isolates, while the number of isolates from the yellow corn source was only four. Interestingly, the following anti-fungal has an impact effect against isolate Azole resistance A. flavus: the inhibition zone of the following Ketoconazole> Voricazole, >Terbinfine was 40 mm, 20 mm, and 12 mm, respectively, while the rest of the anti-fungal agents have no effect against Azole resistance A. flavus. Similarly, the inhibition zones of the following antifungal agents were as follows: 41 mm for Terbinfine, 13 mm for Voricazole, and 11 mm for Ketoconazole against Aspergillus flavus that was isolated from yellow corn. The physiochemical characterization of zinc oxide nanoparticles provides evidence that ZnO-NPs associate with Chlorella vulgaris and have been fabricated by the precipitation method with a diameter of 25 nm. Then, we applied the zinc oxide nanoparticle against isolate Azole resistance A. flavus and the data indicate that the anti-fungal activity of ZnO-NPs has an impact effect against isolate Azole resistance A. flavus. The inhibition zone of zinc oxide nanoparticles against A. flavus (that was isolated from white corn was 50 mm with MIC equal 50 mg/ml while, The inhibition zone of zinc oxide nanoparticle against Azole resistance A. flavus isolated from yellow corn was 14 nm and MIC equal 25 mg/ml which indicated that zinc oxide nanoparticle   give a better result against Azole resistance A. flavus isolated from maize .

  1. Introduction is very hectic and does not properly introduce the main classes of fungicides or the specific fungicide azole, central to this study, the importance of different types of maize nor the basis for selecting ZnO nanoparticles for the study

AUTHORS: Thank you very much for your valuable suggestion. Thanks for reviewer’s comments; we rewrite the introduction again.

Plant pathogens may be lead to reduction of the yield and quality of agricultural products and may cause substantial economic losses. The latter affect food security at household, national and global levels [4]. The reporting of yield losses varies significantly depending on region and target crops. Previous studies reported that crop losses for five of the commodities most widely grown: Wheat, rice, maize, potatoes and soybean, were in the range of 20 to 30 % Food security and ‘Zero Hunger’ are among the global goals declared by the United Nations [5]. Efforts to reduce fungal diseases are crucial for guaranteeing sufficient high quality food for a growing world population. Modern agriculture has been increasingly reliant on the application of fungicides to mitigate fungi-related crop losses and fulfill this goal. Fungicides include a broad range of compounds with different modes of action belonging to various chemical classes.. The extensive use of chemical pesticides in agriculture also raises concerns for public health. Exposure experiments in rats showed endocrine-disrupting, biochemical, histopathological, and hematological effects [6]. Several fungicides are characterized as hazardous chemicals by the World Health Organization (WHO) and are banned in the European Union. The use of azoles for non-medical purposes has a wide span from agriculture and horticulture to the prevention of post-harvest losses [7]. Azoles target a broad spectrum of fungal pathogens in various crops. Aspergillus flavus is a plant pathogen that is able to infect and colonize many crops, such as corn, cotton. It causes an economic problem due to the infection of strategic crops like maize [4]. A. flavus causes subsequent contamination of the grain with the fungal metabolite aflatoxin. Aflatoxins classify as polyketide-derived furanocoumarins are produced by the genus Aspergillus and are classified as highly carcinogenic and toxic materials for humans and livestock [5-6]. Maize is an economic crop that provides nutrients for humans and livestock. It cultivates in many tropical areas, such as North America, Africa, and the Middle East. It includes three types (red, yellow, and white) [5]. Nanotechnology contributes in many different fields, such as medicine, industry, and agriculture. Nowadays, researchers are focusing on the benefits of nanomaterials such as their large surfaces and their optical and chemical properties. Nanomaterials consist of many branches, such as noble metals and metal oxides [6]. Zinc oxide has been applied in the pharmaceutical and cosmetic industries in powder and ointment forms. Also, it has been successful as an antimicrobial, antifungal, and anti-cancer drug [8]. During our work, we isolated A. flavus from 100 samples of maize grains (50 samples from white maize and 50 samples from yellow maize) and then investigated the triazole susceptibility of isolates against antifungal drugs, determined the MIC of the collected isolates, and finally, utilized zinc oxide nanoparticles as fungicides against Azole-resistant A. flavus isolates.

  1. The construction of sentences in Materials and Methods section is not correct and the text is very difficult to understand

 AUTHORS: Thank you very much for your valuable suggestion. Thanks for reviewer’s comments; we rewrite the Material and Methods  section again.

Antifungal disk diffusion method

The fungal strains were cultured on Sabouraud dextrose agar and incubated at 35°C for 24 hours and for 5 days on potato dextrose agar slant for the mold fungi. Using a sterile loop, pure colonies of the Aspergillus flavus species were transferred into a tube containing sterile normal saline. For the mold, 1 mL of sterile distilled water supplemented with 0.1% Tween 20 was used to cover and resuspend the colonies. Using a hemocytometer, the suspension was adjusted to 2–5×106 conidia/mL. The suspension was further diluted 1:10 to obtain final working inoculums 2–5×105 conidia/mL. The inoculums were poured over MHA supplemented with 2% of glucose. The sterile 6 mm disks that were impregnated with 20 μL test compound (with a concentration of 10 mg/mL) were placed over the plate.  Standard antifungal drug Nystatin was used as positive control sterile distilled water as negative control and incubated at 35°C for 48 hours. The zone of inhibition was measured in millimeter [11].

The susceptibility of Aspergillus flavus isolates

The MICs for Aspergillus  flavus were determined using the reference procedure of the Antifungal Suscep¬tibility Testing of CLSI M27-A3 and EUCAST for the testing of fermentative yeasts. MICs for Aspergillus  flavus were determined in accordance with EUCAST and CLSI M38-A. Briefly, testing was performed in sterile 96-well microtiter plates with Roswell Park Memorial Institute (RPMI) 1640 medium with l-glutamine, without sodium bicarbonate (NaHCO3, RPMI 1640; Gibco, Carlsbad, CA, USA) supplemented with 2% glucose, buffered to pH 7.0 with 4-(2-hydroxyethyl)-1-piperazineethanesulfonic acid (HEPES) medium[12-14].

Preparation of fungal inoculums

Regarding preparation of Aspergillus  flavus inoculums, the fungal strains were subcultured on Sabouraud’s dextrose agar slant and incubated for 24–48 hours at 35°C to obtain a freshly grown pure culture. Homogenous suspension was adjusted to 0.5 McFarland standards. Then, the inoculum size was further adjusted to 0.5×105 or 2.5×105. In addition, the mold suspension of conidia was obtained from 5 days culture Sabor and dextrose agar slant incubated at 35°C. Colonies were covered with 5 mL of sterile distilled water supplemented with Tween 20. The conidia were collected with a sterile cotton swab, transferred to a sterile tube, and vortexed to homogenize the suspension. The suspension was standardized by counting the conidia in a hemocytometer to 2–5×106 conidia/mL. The sus¬pension was diluted 1:10 with RPMI to obtain final inoculums of 2–5×105 conidia/mL. A total of 50 μL of each compound concentration and 50 μL of fungal suspension were added to each well for the negative control lane, but 100 μL of broth was added to the positive control lane (each well reached the final desired concentration of 2–5×105 CFU/mL). The plate was sealed with aluminum foil and incubated at 35°C for 2,448 hours in humid atmosphere. The MIC was determined using an ELISA reader at 530 nm for the yeast species and visually for mold species after 48 hours of incubation as the lowest concentration of drug that resulted in 50% inhibition of growth compared to that drug-free growth control.

  1. It is unclear what is shown on Figures 1 and 2..

AUTHORS: Thank you very much for your valuable suggestion. Thanks for reviewer’s comments: we do all your request

Figure 3. FTIR image of zinc oxide nanoparticles

Figure 4. XRD image of zinc oxide nanoparticles

  1. In Materials and Methods, it is not totally clear how ZOI and MIC were determined and measured. Why were ZnO particles dissolved in DMSO? Was DMSO used as a negative control

AUTHORS: Thank you very much for your valuable suggestion. Thanks for reviewer’s comments, ZnO dissolve (partial ) in DMSO , Also, DMSO used as used as a negative control.

  1. It is totally unclear how the authors find based on Figure 5 that the size of ZnO particles was 25 nm. Scale bar on SEM image is 3 micrometers, so the ZnO particles look more around 500-1000 nm in diameter. The TEM image on Figure 6 is of relatively bad quality and indicates that some of the larger particulates are indeed aggregates of ca 25 nm sized nanoparticles.

AUTHORS: Thank you very much for your valuable suggestion. Thanks for reviewer’s comments, we insert anther images first one is SEM image with 1 µm and the second for TEM with 100 nm scale

In Table 4, MIC is given in mg, is it mg/L, mg/ml or mg of something else? How could ZOI be measured in nanometers? (e.g, 15 nm=0.015 micrometers=0.000015 millimeters; how can this be measured?) Have the authors calculated any standard deviations?

AUTHORS: Thank you very much for your valuable suggestion. Thanks for reviewer’s comments, we use  mg/ml

Reviewer 2 Report

Dear Editor,

The authors have reported the ability of zinc oxide nanoparticle to act as antifungal agents against Azole resistance Aspergillus flavus isolated from yellow and white maize. The current study is interesting. However, to meet the journal's quality standards, the following minor comments need to be addressed.

1- There are many grammatical errors in the manuscript. Please review the entire manuscript.

2. The Introduction can be polished and improved. The novelty problem statement described by the authors should be emphasized to attract general readers. Also authors should elaborate the general applicability of the current work.

3. Authors may rearrange/polish the text and elaborated “Material and Methods” section the way so anybody can repeat the procedures, like a recipe. If there is process flow diagram can be added for sampling, isolation and identification of Aspergillus flavus and also for ZnO synthesis, it would be helpful to non-specialist readers.

4-The font size of the different groups on the FTIR figure (Figure 3) is very small and needed to be enlarged.

5- It is recommended to incorporate Miller indices on X-Ray diffraction peaks (Figure 4) for crystal identification.

Author Response

 POINTWISE RESPONSE TO REVIEWERS’S COMMENTS ON Molecule –MDPI

Dear Molecule –MDPI Editor, Dear Reviewers,

Thank you for the reviewer comments for molecule –MDPI; they are very helpful to improve our manuscript. Please find attached the revised article entitled “Fungicidal activity of zinc oxide nanoparticles against Azole-Resistant Aspergillus  flavus isolated from yellow and white maize ”, for consideration for publication in Molecules- MDPI.

Each reviewer comment is indicated below, together with the responses (clarifications/ revisions) we have included in the revision.

It is hoped that this revision satisfies the queries raised by respected reviewers and manuscript will be considered for publication in Molecules- MDPI.

With kind regards

Dr. Amr Hassan

Reviewer-2

.

  1. There are many grammatical errors in the manuscript. Please review the entire manuscript.

AUTHORS: Thank you very much for your valuable suggestion. Thanks for reviewer’s comments; we fixed all the errors.

  1. The Introduction can be polished and improved. The novelty problem statement described by the authors should be emphasized to attract general readers. Also authors should elaborate the general applicability of the current work

AUTHORS: Thank you very much for your valuable suggestion. Thanks for reviewer’s comments; we rewrite the abstract and introduction again.

Plant pathogens may be lead to reduction of the yield and quality of agricultural products and may cause substantial economic losses. The latter affect food security at household, national and global levels [4]. The reporting of yield losses varies significantly depending on region and target crops. Previous studies reported that crop losses for five of the commodities most widely grown: Wheat, rice, maize, potatoes and soybean, were in the range of 20 to 30 % Food security and ‘Zero Hunger’ are among the global goals declared by the United Nations [5]. Efforts to reduce fungal diseases are crucial for guaranteeing sufficient high quality food for a growing world population. Modern agriculture has been increasingly reliant on the application of fungicides to mitigate fungi-related crop losses and fulfill this goal. Fungicides include a broad range of compounds with different modes of action belonging to various chemical classes.. The extensive use of chemical pesticides in agriculture also raises concerns for public health. Exposure experiments in rats showed endocrine-disrupting, biochemical, histopathological, and hematological effects [6]. Several fungicides are characterized as hazardous chemicals by the World Health Organization (WHO) and are banned in the European Union. The use of azoles for non-medical purposes has a wide span from agriculture and horticulture to the prevention of post-harvest losses [7]. Azoles target a broad spectrum of fungal pathogens in various crops. Aspergillus flavus is a plant pathogen that is able to infect and colonize many crops, such as corn, cotton. It causes an economic problem due to the infection of strategic crops like maize [4]. A. flavus causes subsequent contamination of the grain with the fungal metabolite aflatoxin. Aflatoxins classify as polyketide-derived furanocoumarins are produced by the genus Aspergillus and are classified as highly carcinogenic and toxic materials for humans and livestock [5-6]. Maize is an economic crop that provides nutrients for humans and livestock. It cultivates in many tropical areas, such as North America, Africa, and the Middle East. It includes three types (red, yellow, and white) [5]. Nanotechnology contributes in many different fields, such as medicine, industry, and agriculture. Nowadays, researchers are focusing on the benefits of nanomaterials such as their large surfaces and their optical and chemical properties. Nanomaterials consist of many branches, such as noble metals and metal oxides [6]. Zinc oxide has been applied in the pharmaceutical and cosmetic industries in powder and ointment forms. Also, it has been successful as an antimicrobial, antifungal, and anti-cancer drug [8]. During our work, we isolated A. flavus from 100 samples of maize grains (50 samples from white maize and 50 samples from yellow maize) and then investigated the triazole susceptibility of isolates against antifungal drugs, determined the MIC of the collected isolates, and finally, utilized zinc oxide nanoparticles as fungicides against Azole-resistant A. flavus isolates.

  1. Authors may rearrange/polish the text and elaborated “Material and Methods” section the way so anybody can repeat the procedures, like a recipe. If there is process flow diagram can be added for sampling, isolation and identification of Aspergillus flavus and also for ZnO synthesis, it would be helpful to non-specialist readers.

 AUTHORS: Thank you very much for your valuable suggestion. Thanks for reviewer’s comments; we rewrite the Material and Methods  section again.

Antifungal disk diffusion method

The fungal strains were cultured on Sabouraud dextrose agar and incubated at 35°C for 24 hours and for 5 days on potato dextrose agar slant for the mold fungi. Using a sterile loop, pure colonies of the Aspergillus flavus species were transferred into a tube containing sterile normal saline. For the mold, 1 mL of sterile distilled water supplemented with 0.1% Tween 20 was used to cover and resuspend the colonies. Using a hemocytometer, the suspension was adjusted to 2–5×106 conidia/mL. The suspension was further diluted 1:10 to obtain final working inoculums 2–5×105 conidia/mL. The inoculums were poured over MHA supplemented with 2% of glucose. The sterile 6 mm disks that were impregnated with 20 μL test compound (with a concentration of 10 mg/mL) were placed over the plate.  Standard antifungal drug Nystatin was used as positive control sterile distilled water as negative control and incubated at 35°C for 48 hours. The zone of inhibition was measured in millimeter [11].

The susceptibility of Aspergillus flavus isolates

The MICs for Aspergillus  flavus were determined using the reference procedure of the Antifungal Suscep¬tibility Testing of CLSI M27-A3 and EUCAST for the testing of fermentative yeasts. MICs for Aspergillus  flavus were determined in accordance with EUCAST and CLSI M38-A. Briefly, testing was performed in sterile 96-well microtiter plates with Roswell Park Memorial Institute (RPMI) 1640 medium with l-glutamine, without sodium bicarbonate (NaHCO3, RPMI 1640; Gibco, Carlsbad, CA, USA) supplemented with 2% glucose, buffered to pH 7.0 with 4-(2-hydroxyethyl)-1-piperazineethanesulfonic acid (HEPES) medium[12-14].

Preparation of fungal inoculums

Regarding preparation of Aspergillus  flavus inoculums, the fungal strains were subcultured on Sabouraud’s dextrose agar slant and incubated for 24–48 hours at 35°C to obtain a freshly grown pure culture. Homogenous suspension was adjusted to 0.5 McFarland standards. Then, the inoculum size was further adjusted to 0.5×105 or 2.5×105. In addition, the mold suspension of conidia was obtained from 5 days culture Sabor and dextrose agar slant incubated at 35°C. Colonies were covered with 5 mL of sterile distilled water supplemented with Tween 20. The conidia were collected with a sterile cotton swab, transferred to a sterile tube, and vortexed to homogenize the suspension. The suspension was standardized by counting the conidia in a hemocytometer to 2–5×106 conidia/mL. The sus¬pension was diluted 1:10 with RPMI to obtain final inoculums of 2–5×105 conidia/mL. A total of 50 μL of each compound concentration and 50 μL of fungal suspension were added to each well for the negative control lane, but 100 μL of broth was added to the positive control lane (each well reached the final desired concentration of 2–5×105 CFU/mL). The plate was sealed with aluminum foil and incubated at 35°C for 2,448 hours in humid atmosphere. The MIC was determined using an ELISA reader at 530 nm for the yeast species and visually for mold species after 48 hours of incubation as the lowest concentration of drug that resulted in 50% inhibition of growth compared to that drug-free growth control.

  1. The font size of the different groups on the FTIR figure (Figure 3) is very small and needed to be enlarged.

AUTHORS: Thank you very much for your valuable suggestion. Thanks for reviewer’s comments: we do all your request

Figure 3. FTIR image of zinc oxide nanoparticles

  1. It is recommended to incorporate Miller indices on X-Ray diffraction peaks (Figure 4) for crystal identification

AUTHORS: Thank you very much for your valuable suggestion. Thanks for reviewer’s comments: we do your entire request

Figure 4. XRD image of zinc oxide nanoparticles

Reviewer 3 Report

Please do the extensive english correction. Improve the entire manuscript. Please give a scientific explanation on why your nanoparticle seemed more effective against the fungus?

Author Response

 POINTWISE RESPONSE TO REVIEWERS’S COMMENTS ON Molecule –MDPI

Dear Molecule –MDPI Editor, Dear Reviewers,

Thank you for the reviewer comments for molecule –MDPI; they are very helpful to improve our manuscript. Please find attached the revised article entitled “Fungicidal activity of zinc oxide nanoparticles against Azole-Resistant Aspergillus  flavus isolated from yellow and white maize ”, for consideration for publication in Molecules- MDPI.

Each reviewer comment is indicated below, together with the responses (clarifications/ revisions) we have included in the revision.

It is hoped that this revision satisfies the queries raised by respected reviewers and manuscript will be considered for publication in Molecules- MDPI.

With kind regards

Dr. Amr Hassan

Reviewer-3

.

  1. Please do the extensive English correction. Improve the entire manuscript. Please give a scientific explanation on why your nanoparticle seemed more effective against the fungus?

 AUTHORS: Thank you very much for your valuable suggestion. Thanks for reviewer’s comments; we fixed all the errors. Also, we give a scientific explanation on why your nanoparticle seemed more effective against the fungus

The anti-fungal mechanism of zinc oxide nanoparticles depend upon on size and shape [25]. Nanoparticles exist in a different shape such as rod shaped or spherical [26-28], which effect on the surface area to volume ratio and the ability to cause physical damage to cells. The Proposed fungicidal mechanisms of nanoparticles are extrapolated from mechanisms proposed for bacteria and include interaction with thiol groups of vital enzymes leading to enzyme inactivation [29], and killing by oxidative stress [25,30]. ROS was, however, demonstrated to cause membrane and cell wall damage [31]. Hence, scientists noticed that the addition of nanoparticles can be inducing the generation of ROS in time or dose – dependent manner [31].

Round 2

Reviewer 3 Report

Comments is satisfactorily addressed. Please improve your flow of language. Other things are up to the mark.

Author Response

Dear reviewer, 
I'd like to thank you for your time and effort in revising and editing the manuscript until it was as accurate as possible. I worked on all the grammatical errors and fixed them. The manuscript was then sent to a friend who worked as Professor  in the United States for grammar review. He will send it back quickly, and I will submit it.
Best regards
Dr. Eman Sharaf